# Deterioration of Mortar Bars Using Binary and Ternary Mixtures Immersed in Sodium Sulfate Solutions

**Federico Aguayo [1,*] and Mehrab Nodehi [2]**

1 College of Built Environments, University of Washington, Seattle, WA 98105, USA
2 Department of Civil Engineering, University of California, Davis, CA 95616, USA
* Correspondence: aguayo12@uw.edu

**Abstract:** In this study, the performance of several binary and ternary mixtures containing high-calcium fly ash and other pozzolans, such as Class F fly ash and silica fume, were investigated for their sulfate resistance using different sodium sulfate solutions. The mortar bars were placed in a similar sulfate solution as per modified ASTM C 1012/1012M (33,800 ppm $SO_4^{2-}$) with a less severe sulfate solution (6000 ppm $SO_4^{2-}$) has been tested to resemble actual field performance for a duration of 18 months. The phase composition of the mortar samples was investigated using X-ray diffraction and scanning electron microscope coupled with energy dispersive spectroscopy (SEM/EDS). Results show that the mortar bars placed in the moderate sulfate concentration experience less expansion and deterioration than the same bars placed in the higher sulfate concentration. Storage in sodium sulfate solutions resulted in the formation of ettringite and gypsum in both sulfate concentrations. Replacement of cement by high-calcium fly ash showed significantly higher amounts of ettringite formation, especially for the mortar bars stored in the higher sulfate concentration. SEM analysis revealed ettringite to be the primary cause of disruption and deterioration observed in the mortar bars.

**Keywords:** sulfate attack; sulfate concentration; ettringite; gypsum; expansion





## 1. Introduction

Climate change, rapid urbanization and an increase in population, has resulted in a considerable social reliance on the aging infrastructure to keep up with the increasing demand of utilizing natural resources. In the construction sector, a large proportion of key infrastructures are built with cementitious composites that are known to be vulnerable to the deteriorating factors, such as the environmental exposure, loading conditions and chemical agents [1–3]. In this regard, one of the key components of such physico-durability concern is that of sulfate attack that can result in expansion, cracking and spalling of concrete structures and lead to a loss of physico-durability properties. Most commonly, sulfate exposure takes place through external sources, such as soil, ground and even sea water, diffusing through the concrete pores. This process is known to result in a series of expansive reactions of sulfate ions with aluminum containing phases and/or calcium hydroxide to form ettringite or gypsum, respectively. The formation of these products can lead to volumetric expansion and cracking of the surface layer, thus, easing further penetration of sulfate ions into the concrete and resulting in more severe damage [4].

To evaluate this property in cementitious composites, the most commonly recommended methods of evaluating the sulfate resistance has been to test specimens under exposure to natural sulfate conditions in the field while being periodically examined over time. However, unless the concrete is very porous, field exposure cannot provide relatively rapid results and there are many uncontrolled factors, such as temperature and humidity that can considerably affect this testing process. Generally, ASTM C 1012 [5] is commonly used to determine the performance of various mortar mixtures under sulfate attack in controlled laboratory conditions. In this method, standard mortars are cast and monitored for

their expansion periodically while submerged in a 5% sodium sulfate solution for up to 18 months. The sulfate concentration specified in the test solution, however, has been a subject to numerous research studies for being considered far too aggressive and not realistic of the actual field conditions. Moreover, testing with ASTM C 1012 [5] has been reported as having poor correlation to field performance [5–7] that requires considerable time to qualify performance, making it a very unpopular test among researchers and practitioners.

The advantage of obtaining a test result in a relatively short period of time (when compared to the service life of a structure) is typically accompanied by changes in the process of deterioration and has to be brought into question [8]. Some of the early research investigating the mechanisms of sulfate of attack found that with increasing sulfate concentration, gypsum is the main phase present [9–11], whereas field studies have shown ettringite formation to be the primary phase present and main cause of deterioration in concrete structures [12]. In many cases, the source and type of sulfate (i.e., $Na^+$, $Mg^{2+}$, $Ca^{2+}$) can also dictate the type of chemical reaction that may occur in the field [9]. Testing with magnesium sulfate, for instance, is reported as being less expansive than sodium sulfate and depending on the binder composition, may result in surface deterioration rather than expansion [13–15]. However, many studies also report the opposite with magnesium being the most aggressive due to the formation of a corrosive magnesium-silicate-hydrate (M-S-H) and commonly associated with loss of binding properties rather than expansion [8,16]. Although calcium sulfate (gypsum) is a common source of sulfate in soils, it has not been studied as much in the laboratory and has not caused as much damage in the field, presumably due to its lower solubility in comparison to magnesium and sodium sulfate [9].

In this regard, the present study evaluates the influence of the sulfate concentration in the test solutions on the formation of sulfate products, particularly ettringite and gypsum. To accurately test the samples, mortar bars were placed in two sodium sulfate solutions (5.0% and 0.89% $Na_2SO_4$) and evaluated for their length change for 18 months. As a comparison, the performance was compared to that of standard mortar bars cast as per ASTM C 1012 [5]. In addition, X-ray diffraction (XRD) and scanning electron microscopy (SEM) coupled with energy dispersive spectroscopy (EDS) was used to quantify and characterize the phases present in the system at different intervals.

## 2. Research Significance

Many studies have used the accelerated mortar bar method ASTM C 1012/1012M [5], in combination with ACI 201.2R-08 [17], to investigate the sulfate resistance of mortar mixtures, especially when testing fly ashes with calcium contents of more than 20% CaO [17]. In the North America where sulfate exposure conditions are encountered, the sulfate levels are often less aggressive than that used in the accelerated mortar bar method. This raises significant controversy regarding the underlying mechanisms and whether the method provides a direct comparison to field performance. As a result, this study provides an evaluation of mortar bar using similar methods as per ASTM C 1012/1012M [5] but investigate their performance using a less aggressive sulfate solution and draw a more realistic comparison between this test method and those of the actual field exposures.

## 3. Materials and Experimental Procedures

In this study, mortar bars were fully submerged in moderate and aggressive sulfate concentration and periodically monitored for their length change over 18 months of time period. Scanning electron microscopy (SEM), energy-dispersive spectroscopy (EDS), and X-ray diffraction (XRD) were used to identify and characterize the microstructure formed after sulfate exposure.

### 3.1. Materials

Two portland cements were procured from within the state of Texas for making the mortar, a Type I cement (C1) with a high $C_3A$ content of 10% and a moderate sulfate

resistant Type I/II cement (C2) with a $C_3A$ content of 7%. These cements were designated as the controls and were assumed to have poor and moderate resistance to external sulfate attack based on their $C_3A$ content as prescribed in ASTM C 150 [18]. Further, fine aggregates conforming to ASTM C778 [19] standard graded Ottawa Sand from Humboldt Inc. [20] has been used.

To evaluate the influence of sulfate concentration on binary and ternary mixtures, a wide range of supplementary cementitious materials were chosen for this study. A high-calcium (HC) fly ash and low calcium (LC) fly ash were used in combination with both portland cements at 30 and 25% and replacement by mass of cement, respectively. Additionally, 5% silica fume (SF) was used as part of a ternary blend with 35% HC fly ash. The HC fly ash had a CaO = 28.98% and is known to be susceptible to sulfate attack [6,7,21–23]. The chemical composition of the cementitious materials, as well as the phase composition of the cements used is presented in Tables 1 and 2, respectively.

**Table 1.** Chemical compositions of cementitious materials (% by mass).

| Cement Type | $SiO_2$ | $Al_2O_3$ | $Fe_2O_3$ | CaO | MgO | $Na_2O$ | $K_2O$ | $SO_3$ | LOI |
|---|---|---|---|---|---|---|---|---|---|
| Type I (C1) | 20.36 | 5.43 | 2.50 | 63.12 | 1.35 | 0.09 | 1.03 | 3.23 | 2.60 |
| Type I/II (C2) | 20.38 | 4.90 | 3.55 | 63.62 | 1.14 | 0.11 | 0.67 | 2.86 | 2.20 |
| **Supplementary Cementitious Materials** | | | | | | | | | |
| Class C Fly Ash (HC) | 30.76 | 17.75 | 5.98 | 28.98 | 6.55 | 2.15 | 0.3 | 3.64 | - |
| Class F Fly Ash (LC) | 48.48 | 25.01 | 3.56 | 15.92 | 2.5 | 0.3 | 0.71 | 0.72 | - |
| Silica Fume (SF) | 93.17 | - | 2.1 | 0.8 | 0.3 | - | - | 0.2 | - |

**Table 2.** Phase compositions of cements (% by mass) *.

| Cements | $C_3S$ | $C_2S$ | $C_3A$ | $C_4AF$ |
|---|---|---|---|---|
| Type I | 62.12 | 11.52 | 10.16 | 7.61 |
| Type I/II | 66.06 | 8.60 | 6.98 | 10.80 |

* (Bogue calculation).

### 3.2. Methods and Testing Procedures

Length Changes

Mixture proportions are presented in Table 3. A modified version of ASTM C 1012 [5] was used to measure the expansion caused by sulfate attack. The most recent version of ASTM C 1012/1012M [5] specifies the mortar bars and cubes to be stored in a sealed curing container on top of risers above water, and stored in an oven at 35 °C ± 3 °C (95 °F ± 5 °F). However, mortar bars used in this study were cast and cured following the procedures outlined in the 2012 version of ASTM C 1012/1012M [5]. Other modifications include a fixed water-cementitious ratio (*w/cm*) of 0.485 (versus the comparable flow to the control cement mixture), reduced number of bars tested (five versus six), and the addition of a lower sodium sulfate concentration (6000 ppm versus 33,800 ppm $SO_4^{2-}$).

Mortars were mixed according to ASTM C 109 [24]. Mortar bars 25 × 25 × 285 mm (1 × 1 × 11.25 in) and cubes 50 mm (2 in) were cast, sealed in double Ziploc bags and submerged in a curing tank to cure for the first 24 h at 35 °C ± 3 °C (95 °F ± 5 °F). Following the first 24 h of curing, the mortar bars and cubes were then demolded and transferred to a saturated limewater curing tank at 23 °C ± 3 °C (73 °F ± 3 °F). The mortar bars and cubes were allowed to cure until two cubes reached an average compression strength of 20 MPa (2850 psi) or more. Once the strength was achieved, the mortar bars were removed from the limewater, measured according to ASTM C 490 [25] and transferred to a container containing 5% $Na_2SO_4$ (33,800 ppm $SO_4^{2-}$) and 0.89% $Na_2SO_4$ (6000 ppm $SO_4^{2-}$) solution at 23 °C ± 3 °C (73 °F ± 3 °F). Length change was determined in reference to an invar bar before sulfate exposure and after 7, 14, 21, 28, 56, 91, 105, 121, 182 days of exposure and

every 3 months thereafter. Results were based on an average of four specimens. Samples to track the microstructural changes after various immersion periods were taken from the fifth specimen. During each measurement, the solutions were replaced with new 5% and 0.89% sodium sulfate solution at 23 °C ± 2 °C (73 °F ± 3°F) to remove any significant amount of alkalis leaching into the solution and thus increasing the pH.

**Table 3.** Mixture proportions (% by mass).

| Mixture | W/CM | C1 (%) | C2 (%) | HC (%) | LC (%) | SF (%) |
|---------|------|--------|--------|--------|--------|--------|
| C1-Cont | 0.485 | 100 | - | - | - | - |
| C1-30HC | 0.485 | 70 | - | 30 | - | - |
| C1-35HC-5SF | 0.485 | 60 | - | 35 | - | 5 |
| C1-25LC | 0.485 | 75 | - | - | 25 | - |
| C2-Cont | 0.485 | - | 100 | - | - | - |
| C2-30HC | 0.485 | - | 70 | 30 | - | - |
| C2-35HC-5SF | 0.485 | - | 60 | 35 | - | 5 |
| C2-25LC | 0.485 | - | 75 | - | 25 | - |

With SF: silica fume, C1: type I Portland cement, C2: type I/II moderate sulfate resistant Portland cement, with a $C_3A$ content of 7%, HC: high calcium fly ash, LC: low calcium fly ash, and w/cm: water to cement ratio.

*3.3. Scanning Electron Microscopy*

A JEOL JSM-6490LV scanning electron microscope equipped with a secondary electron (SE) detector, back-scattered electron (BSE) detector, and energy-dispersive spectrometer (EDS) was used to observe the microstructural changes of the samples post-sulfate exposure. The samples were examined after 4 months of exposure to sulfate primarily due to the abrupt change in expansion noted between the two different concentrations. Cross-sections of the mortar bar were broken off and sawed down to an 8 mm (0.31 in) thick sample. Samples were prepared by epoxy impregnating, fixed grinding, polishing and placing under vacuum in a desiccator until examined in the SEM. For consistency, the SEM samples are labeled according to their mixture number and exposed sulfate concentration. Mortar bar samples investigated using SEM were not carbon coated. Consequently, the microscope was operated at low vacuum with the pressure varying between 1–10 Pa. In addition, a 15 and 20 kV accelerating voltage (depending on the quality of the image) and 10 mm (0.39 in) working distance was used to optimize the image. EDS analysis was used to identify the elements present in the mortar sample.

*3.4. X-ray Diffraction*

Microstructural changes were also studied using powder collected from mortar bar samples exposed to sodium sulfate solutions. About a 12.5–25 mm (0.5–1.0 in) sample was broken off during periodic measurements and stored under vacuum for no less than three days. Samples were collected and finely ground below 45 micron (0.008 in) before prepping for the XRD. An XRD diffractometer was used to collect the pattern with a position-sensitive detector operating at 40 kV with a 30-mA using a copper target (Cu Kα wavelength 1.54 Å) and a nickel filter and a carbon monochromer. XRD scans were collected from 5–70° 2θ with a step size of 0.2°/min and a dwell time of 6 s. The composition of the hydrated pastes was determined by quantitative X-ray diffraction (Rietveld) analysis on the diffraction patterns. Rietveld analysis is a technique to accurately determine the quantities of crystalline phases present in a sample. The procedure involves determination of calculated XRD patterns through simulation technique based on the structure files for the relevant phases expected within the sample. Simulations were carried out using the TOPAS software. The calculated pattern is refined step-by-step to take into account peak shape, instrumental factors, variations in structures, errors induced during sample preparation, and temperature effects [6,26]. Refinement is a systematic procedure in

Rietveld analysis to obtain the simulated XRD pattern in close agreement with the observed XRD pattern. Results presented in this study are on refinements with a weighted profile R-factor ($R_{wp}$) $\leq$ 10% [27].

## 4. Experimental Results and Discussion

### 4.1. Expansion and Visual Appearance

#### 4.1.1. C1 Mixtures

The accelerated mortar bar expansion results for mixtures using Type I cement at 0.89% and 5.0% sodium sulfate concentrations are presented in Figure 1. As expected, the mixtures placed in the less aggressive sulfate solution exhibited a significantly lower expansion rate compared to their companion bars. During the first four weeks of measurements, only small length changes were observed. Thereafter, mixtures exposed to the more aggressive solutions began to diverge away and experience higher expansions. The fastest expansion was observed for the binary mixture (C1-30HC), which began showing significant expansion at 8 weeks, and ultimately failed after only 15 weeks of exposure in 5% sodium sulfate. The control mixture (C1) observed the second-best performance followed by the ternary (C1-35HC-5SF) and binary (C1-25LC) mixture, which was still measurable after 950 days of exposure.

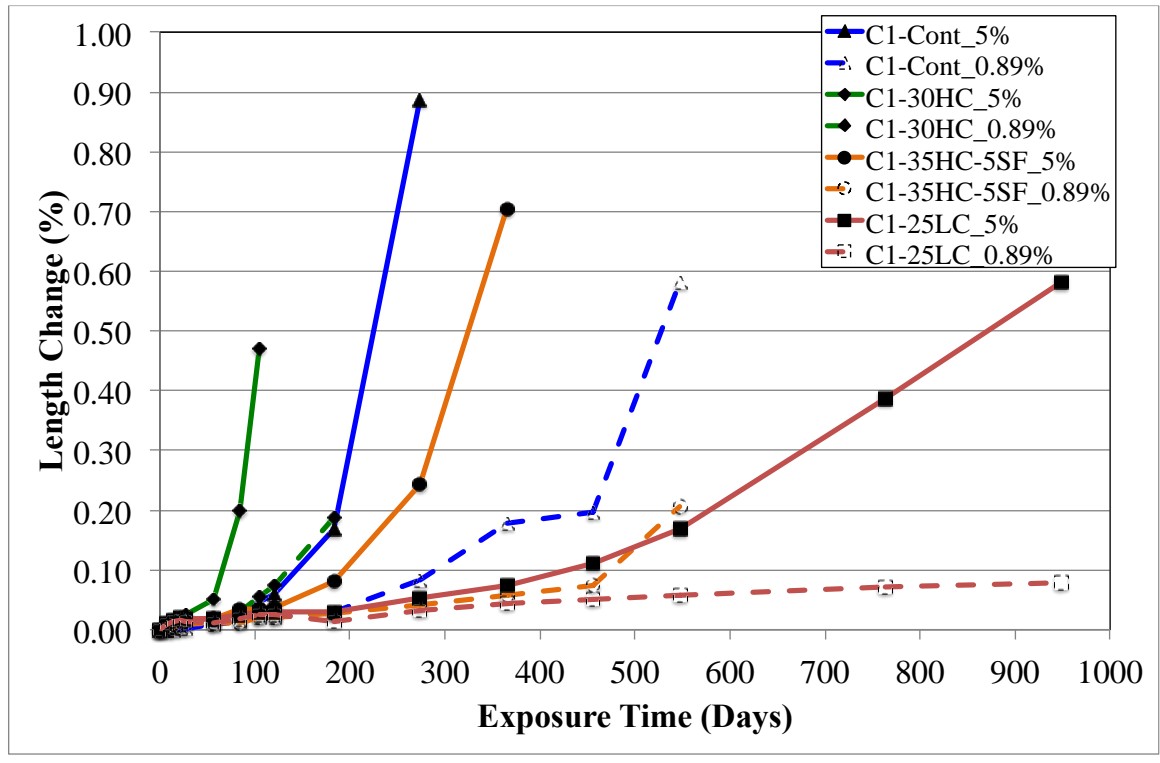

**Figure 1.** Accelerated mortar bar expansion results for Type I mixtures exposed to 5 and 0.89% Na$_2$SO$_4$.

The expansion rates differed significantly between the lower and higher sulfate concentrations. Interestingly, similar performance was observed for those bars placed in 0.89% sodium sulfate showing the same level of performance between the mixtures. With exception to the high-calcium binary mixture, all mortar bars demonstrated delayed expansion in the 0.89% sodium sulfate solution. Mixture C1-30HC exhibited a final expansion value of 0.47% after only 15 weeks of exposure in 5% sodium sulfate and was no longer measurable at 4 months due to complete loss of cohesion of the mortar bars as shown in Figure 2. The bars submerged in the lower concentration had an expansion of 0.18% at 6 months and were still intact with moderate cracking observed, and by 9 months of exposure, the bars observed cracking and surface deterioration, especially at the ends and corners of the bars.

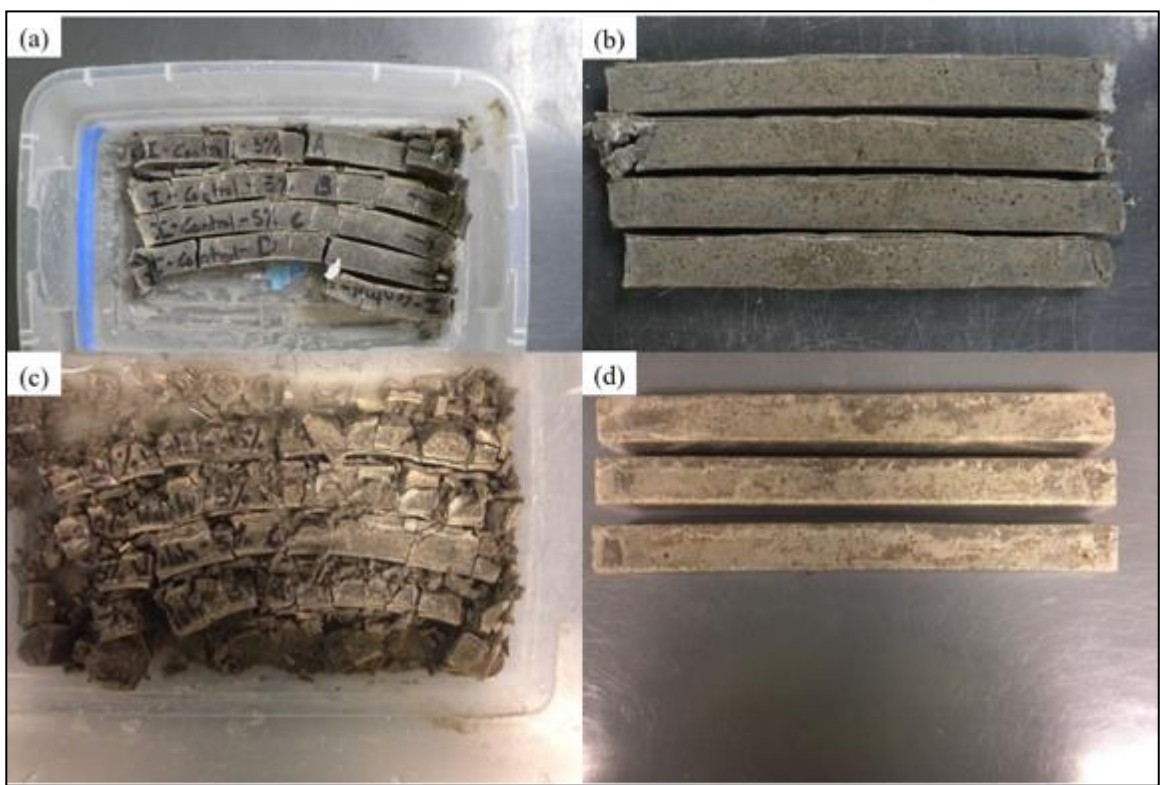

**Figure 2.** Visual appearance in mortar bars showing deterioration in: (**a**) C1 in 5% $Na_2SO_4$ after 1 year; (**b**) C1 in 0.89% after 18 months; (**c**) C1 + 30%HC in 5% $Na_2SO_4$ around 4 months; (**d**) C1 + 30%HC in 0.89% $Na_2SO_4$ at 4 months.

It has been reported by many researchers that the sulfate resistance of high-calcium fly ash mixtures can be improved through small additions of silica fume (3–6%) as a ternary blend [28–30]. Through the addition of silica fume or natural pozzolan, the calcium hydroxide content will decrease and can reduce the severity from gypsum formation. Morever, the pozzolanic reactivity can decrease permeability and ultimately impede the ingress of external sulfates into the paste matrix. Surprisngly, this was not the case for the ternary mixture (C1-35HC-5SF) evalauted in this testing program. Although it exhibited a slower rate of expanson in comparison to the HC binary mixtures, it is clear that the HC fly ash used in this study raises significant concerns with regard to its sulfate performance. Mortar bars placed in 0.89% sodium sulfate showed similar performance with an expansion of 0.21% at 18 months of sulfate exposure.

With exception to the binary LC mixture submerged in 0.89% sodium sulfate solution, all mixures exhibited significant expansion at both concentration. Remarkably, the binary LC mixture did not show any appreciable expansion having only a 0.08% expansion after 950 days of exposure in 0.89% sodium sulfate. Although significant expansion was observed in similar mortar bars placed in 5% sodium sulfate, only moderate cracking and deterioration at the ends and corners were observed on the binary LC mortar bars.

Figure 2 shows the visual appearance of the mortar bars at 12 and 18 months for the control mixture placed in 5% and 0.89% sodium sulfate, respectively, and around 4 months for the binary HC mixture placed in both sulfate solutions. The images to the left show those mixtures placed in 5% sodium sulfate whereas, those on the left were placed in 0.89% solution. From the results showing the visual appearances, two major observations can be drawn:

- The mortar bars submerged in the 5% sodium sulfate solution clearly illustrates the aggressiveness from the concentrated sulfate solution showing severe damage and deterioration in the mortar bars;
- The mortar bars exposed to 0.89% sodium sulfate solution exhibited similar expansion values at later times; however, the noted deterioration is significantly less than those in 5% solution.

These findings demonstrate the severity of performance testing to evaluate durability issues for cementitious mixture. Although there is significant pressure to develop accelerated methods that can provide results in a timely manner, the results may not accurately reflect field-exposed concrete. In many cases, the accelerated method may change the mode of failure and thus, the deterioration that is seen on the sample [31]. The results in Figures 1 and 2 give some indication that this type of phenomenon could be occurring. Moreover, the mode of damage and reduced deterioration observed on the mortar bars submerged in a 'more' realistic sulfate solution may help explain why relatively few cases of sulfate attack are described in the field.

### 4.1.2. C2 Mixtures

The observed length changes for mixtures using C2 (see Figure 3) were smaller than for the previously discussed C1 mixtures. Interestingly, the length changes are very similar for both concentrations during the first 15 weeks (before larger expansions are observed). Thereafter, it appears that cracking of the mortar bars has initiated and the mechanism is controlled by the diffusion-reaction phenomenon as described in [13,32]. The mortar bars evaluated in this study typically showed cracks originating from the finished surface and progressing further towards the center of the bar leading to a warping effect.

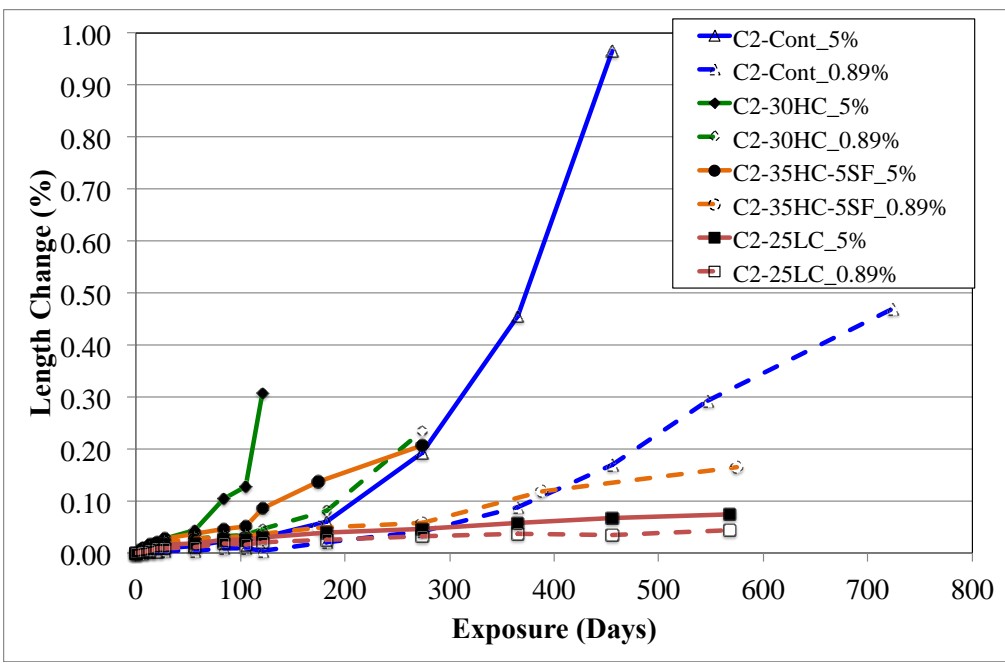

**Figure 3.** Accelerated mortar bar expansion results for Type I/II mixtures exposed to 5 and 0.89% Na$_2$SO$_4$.

As expected, the binary HC mixtures showed poor sulfate resistance showing significant expansion after only 12 weeks in 5% sodium sulfate solution, and after 4 months, the bars showed severe deterioration with most failing at the center of the bar (see Figure 4). The results suggest that the chemical composition of the mixtures prior to sulfate exposure could be affecting the performance. This is likely attributed to the reactive glassy phases

and aluminates available to react in the fly ash mixture thus favoring the formation of ettringite at later ages [22].

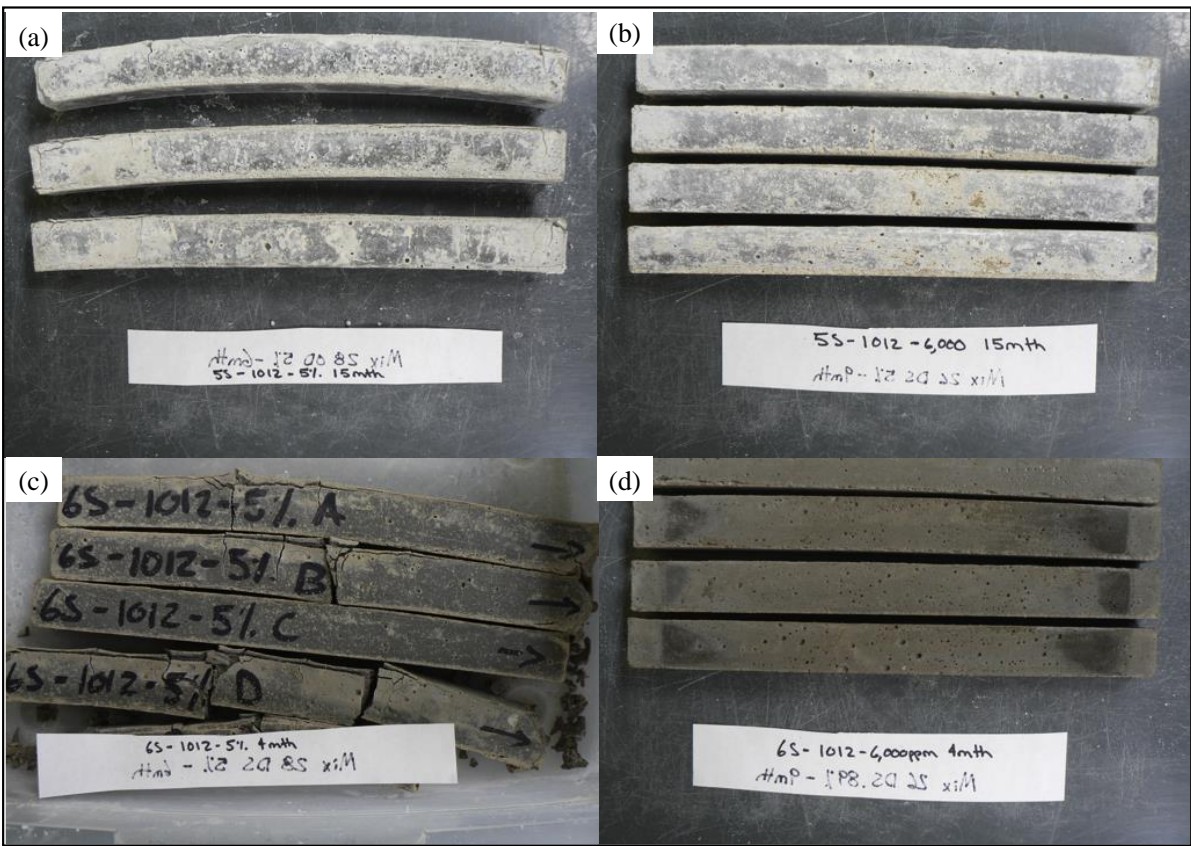

**Figure 4.** Visual appearance of mortar bars showing deterioration in: (**a**) C2 mixture (5S) in 5% $Na_2SO_4$ after 15 months; (**b**) C2 mixture in 0.89% $Na_2SO_4$ after 15 months; (**c**) C2 + 30%HC in 5% $Na_2SO_4$ around 4 months; and (**d**) C1 + 30%HC in 0.89% $Na_2SO_4$ at 4 months.

Santhanam et al. [33] modeled the effects of sodium sulfate concentrations on small mortars. He describes the expansion of mortar in sodium sulfate follows a two-stage process; an initial period of very small expansion until a critical value is achieved, followed by a sudden increase in expansion. As discussed at length by Ref. [34], it can be said that the sulfate attack is a self-accelerating process. Nonehteless, at any sodium sulfate concentration, the duration of the initial stage of expansion is unaffected; however, once the initial level of disruption is achieved, the rate of attack is proportional to the concentration [33]. The findings in this study present similar results for the length change at both sulfate concentrations. The visual presence of moderate size cracks at the corners and finsihed surface of the control (C2) mortar bars seem to be in agreement with the sugggested literature mentioned above (see Figure 4). With exception to the binary HC mixture, all mixtures observed very little change in expansion up to 15 weeks follow by a divergence in expansion rates between both sulfate concentrations.

### 4.2. Microstructural Changes

To observe and identify microstructural changes in the mortar mixture exposed to the two sodium sulfate concentrations, secondary electron (SE) imaging combined with EDS was used to identify elements present in the sample. Additionally, XRD combined with the Rietveld method was performed to quantify the phases present.

Figure 4 months of exposure (Figure 5). The higher concentration appears more distinct and distributed throughout the matrix, which is a consistent with the associated

increase in length change at this period (see Figure 1). Interestingly, ettringite deposits were also evident in the lower sulfate concentration; however, the amount and arrangement were discontinuous and significantly less dense. In comparison to the higher concentration, the images also revealed little to no microcracking within the bulk paste matrix at the lower concentration. The results are consistent with the physical length change observed previously; the mortar bars exhibited minor cracks along the edges and higher expansion values in the higher concentration.

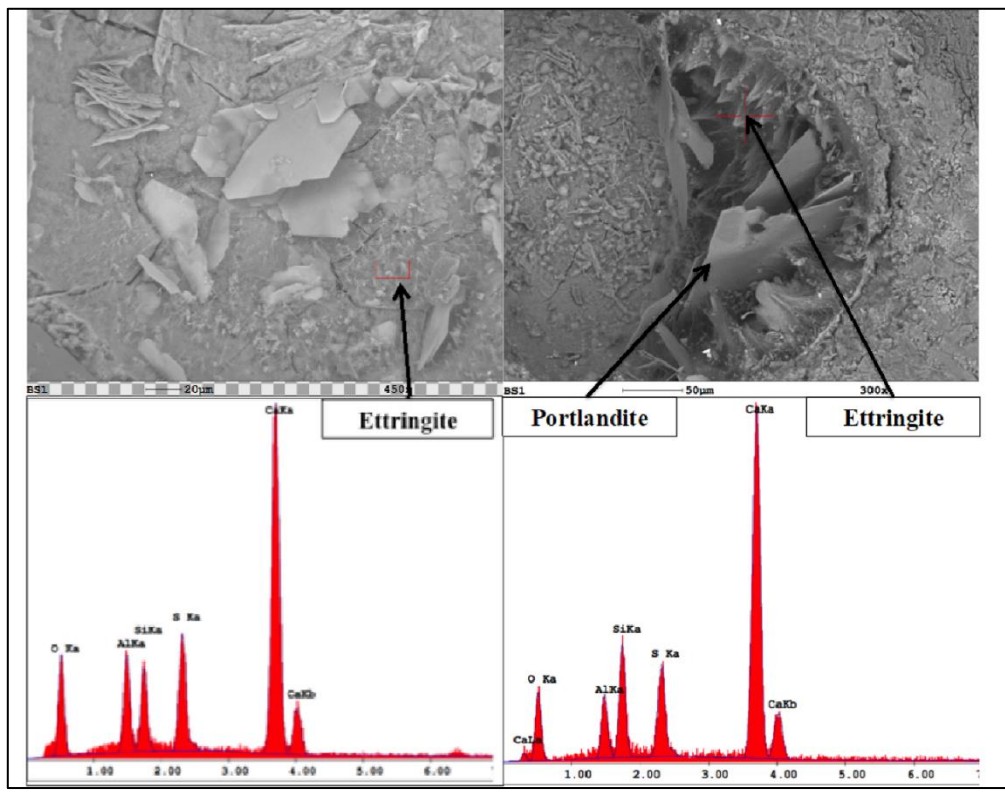

**Figure 5.** SE images coupled with EDS spectrums showing C1 mixture after 4 months exposure in: (**a**) 5% $Na_2SO_4$; and (**b**) 0.89% $Na_2SO_4$.

Further, it is seen that both concentrations showed significant deposits of ettringite in the pores with a remarkably high amount of microcracking in the paste matrix. Similar to the control mixture, the formation of ettringite in the 5% sodium sulfate solution appeared significantly denser and well distributed throughout the system.

It is worth mentioning an interesting observation made in several of the SEM images. In the work presented here, ettringite was commonly found to form in areas near and/or around portlandite crystals in mixtures submerged in the 0.89% sodium sulfate. Figures 5–8 shows several SE images of the binary HC mixture after 4 months of exposure in 0.89% sodium sulfate solution. The figure presents three EDS spectrums illustrating the conversion of portlandite to gypsum, followed by the formation of poorly crystalline ettringite. The above mechanism can be described by the following two chemical reactions between external sodium sulfate solution and calcium hydroxide in the hydrated cement paste [9]:

$$Ca(OH)_2 + Na_2SO_4 + 2H_2O = CaSO_42H_2O + 2NaOH \tag{1}$$

$$Ca_3Al_2O_3(CaSO_4)H_{12} + 2CaSO_42H_2O + 16H_2O = Ca_3Al_2O_3(CaSO_4)_3H_{32} \tag{2}$$

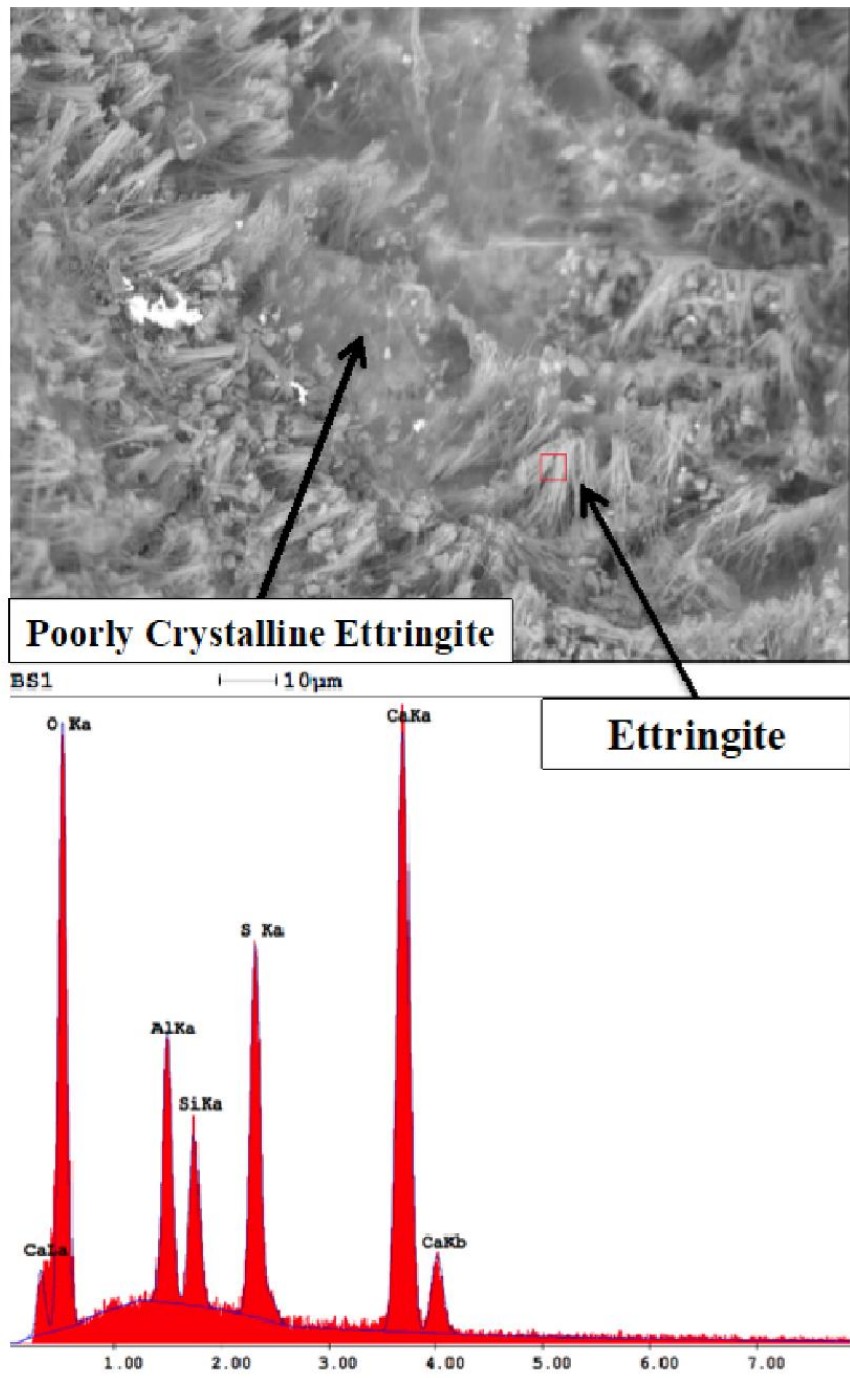

**Figure 6.** SE image coupled with EDS spectrum showing significant ettringite formation in C1-30HC mixture after 4 months exposure in 5% $Na_2SO_4$.

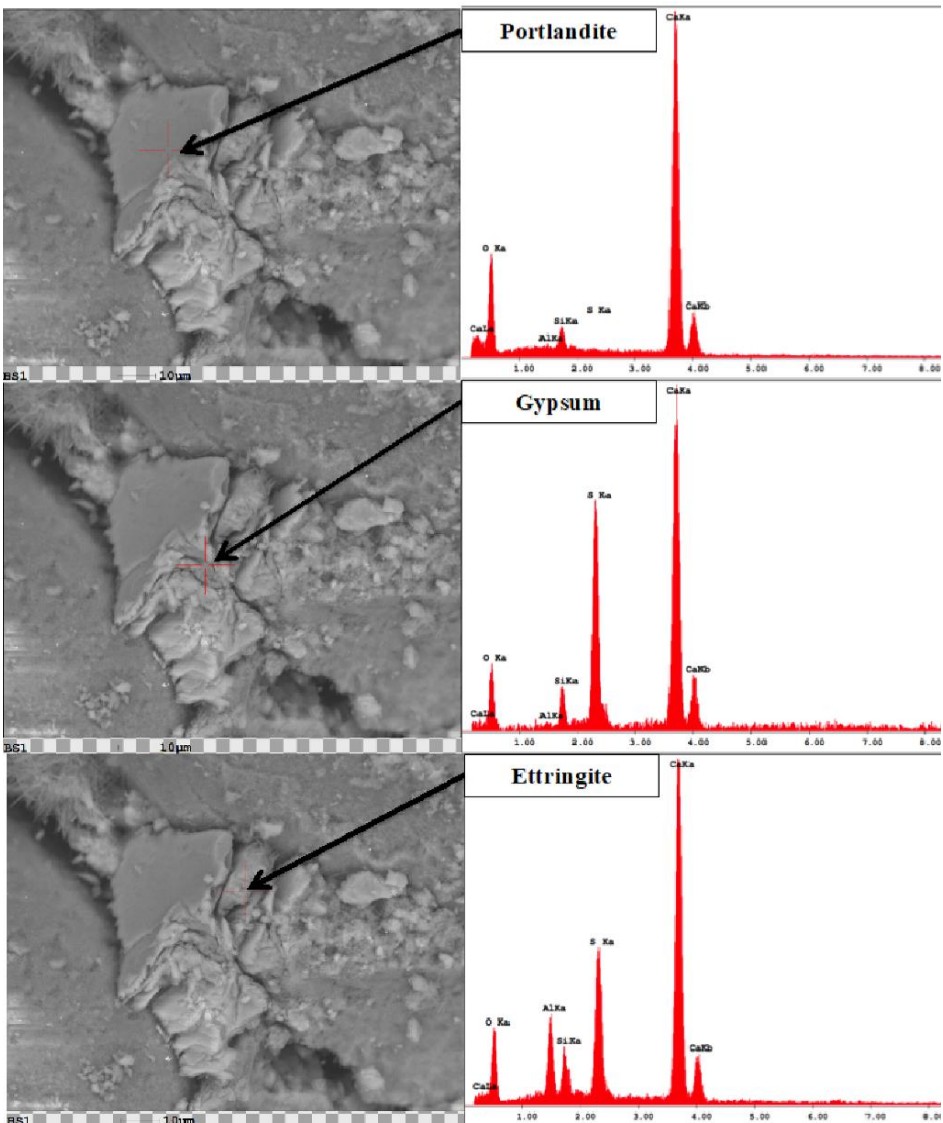

**Figure 7.** SEM images coupled with EDS showing C1-30HC mixture after 4 months exposure in 0.89% $Na_2SO_4$.

The SEM images revealed very few observations of gypsum in any of the mixtures. The main phase present, independent of the solution used was ettringite. Furthermore, in many instances the propagation of cracks was also found in areas where significant ettringite deposits were located in the paste, as shown in Figure 8. According to [35–37], ettringite could only exert enough crystal pressure to cause expansion and cracking in small pores within a certain size range. The SEM images appear to indicate that ettringite crystal growth is the primary cause of expansion even in mixtures submerged in the lower sulfate concentration (0.89% $Na_sSO_4$); the aforementioned mechanism is intensified with increasing concentration and supersaturation of the pores with sulfate ions [33].

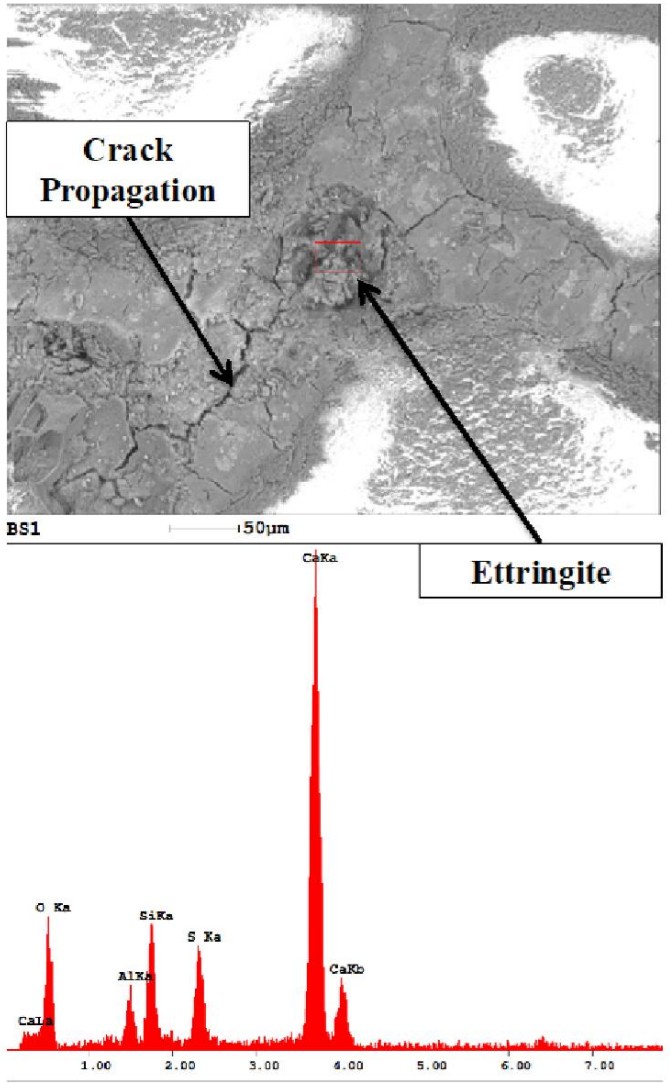

**Figure 8.** SE image of C1-30HC mixture after 4 months exposure in 0.89% Na$_2$SO$_4$.

### 4.3. X-ray Diffraction (XRD)

The phases present in each mixture were identified using XRD after exposure to the sulfate concentrations over time. Figure 9 provides the Rietveld analysis for a diffraction pattern on the control mixture (C1-Cont) after 1 year of exposure in 5% sodium sulfate. The pattern shows the chemical composition in the powder sample in terms of the normalized amounts of 11 crystalline phases present at detectable levels. For all patterns evaluated, monosulfate (M), ettringite (E), gypsum (G), and portlandite (CH) were detected in the qualitative and quantitative Rietveld analysis.

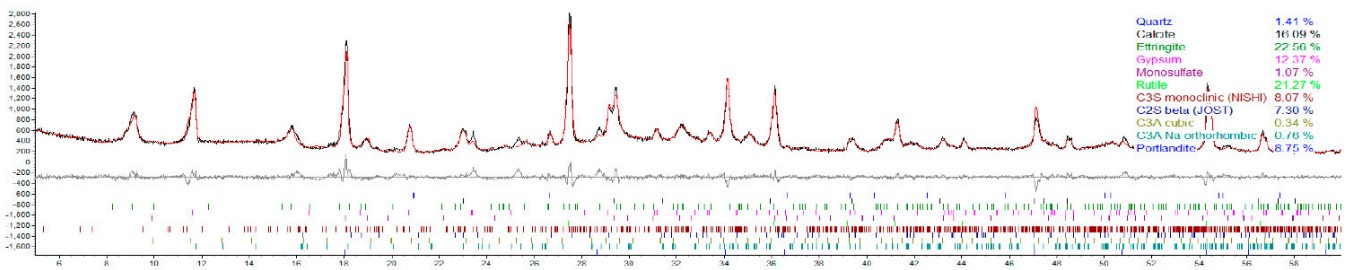

**Figure 9.** Rietveld analysis for control mixture (C1-Cont) after 1 year exposure in 5% Na$_2$SO$_4$.

XRD patterns for the control (C1-Cont) and binary (C1-30HC) mixtures are presented in Figure 10. In both sulfate solutions, the patterns revealed traces of ettringite, gypsum, as well as portlandite. Interestingly, there is a significant drop in the portlandite intensity most likely as a result of gypsum conversion from the external sulfates. The drop is evident along the portlandite peak at about 34.1° 2θ. A significant drop in the portlandite peak is also observed in the 0.89% sodium sulfate solution. Although ettringite is clearly evident in the 5% sodium sulfate patterns, small traces are evident in the 0.89%, which is consistent with the previous SEM results.

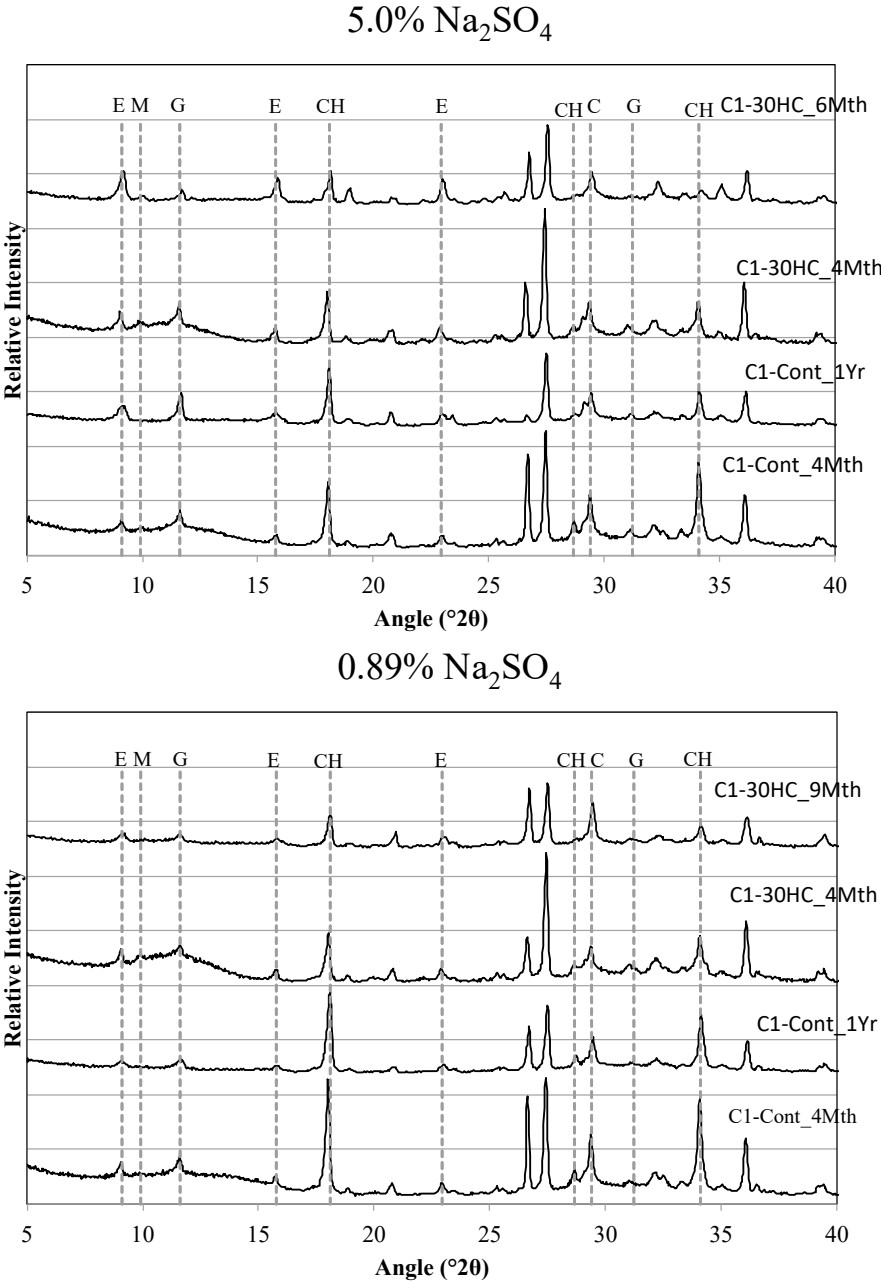

**Figure 10.** XRD traces of control (C1-Cont) and binary (C1-30HC) mixtures.

The percentages of hydration products were determined using rutile (TiO$_2$) as an internal standard and are presented in Figure 11a–c. Ettringite and gypsum phases were present at similar quantities at about 56 days independent of sodium sulfate concentration; however, after one year exposure the ettringite is the dominant phase present with minor

increases in the gypsum phase. Here, the further formation of ettringite leads to faster expansion. This is contradictory to what many researchers have reported. Biczok [38], for instance, reported that the mechanisms of sulfate attack is dependent on the concentration of the sulfate solution where concentration exceeding 8000 ppm $SO_4^{2-}$ (1.2% $Na_2SO_4$), gypsum is the main phase present. According to Müllauer et al. [39], however, ettringite was found to be the primary phase present in higher sulfate concentration (30 g/L $SO_4^{2-}$) and responsible for expansion and damage.

- Ettringite and gypsum phases were also present in mortars immersed in the lower sulfate concentration after one year of exposure; however, ettringite was found in much smaller quantities. At 0.89% sodium sulfate, the control mixture (C1) observed over 70% less ettringite whereas, the binary mixture (C1-30HC) observed over a 140% less ettringite. It is interesting to note the similar ettringite quantities present in the ternary (C1-35HC-5SF) and LC binary (C1-25LC) mixture immersed in the 0.89% sodium sulfate solution. Both mixtures also observed very similar expansion values after one-year exposure. The formation of ettringite does not necessarily result in significant expansion or damage, depending on where and under what conditions it forms [36].

The Rietveld results show that ettringite and gypsum are present at both sulfate concentrations; however, the associated cracking and observed expansion appears to be from the formation of ettringite in small pores. This was also evident in the SEM images previously discussed. Nevertheless, the presence of both may indicate deterioration attributing from both phases.

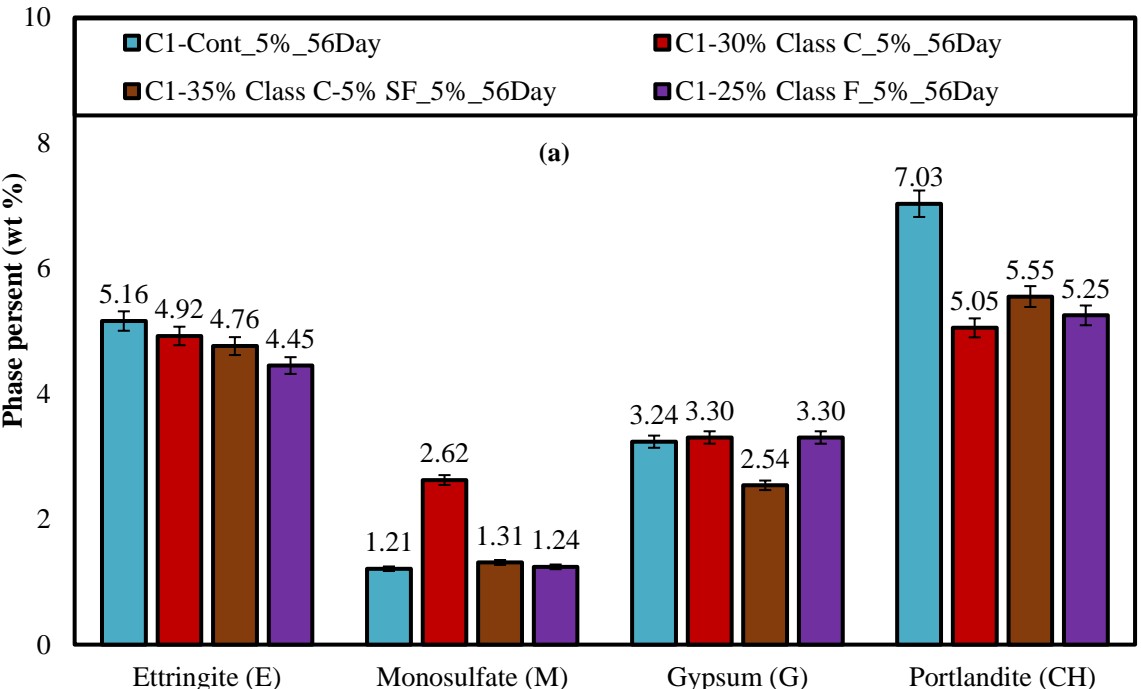

**Figure 11.** *Cont.*

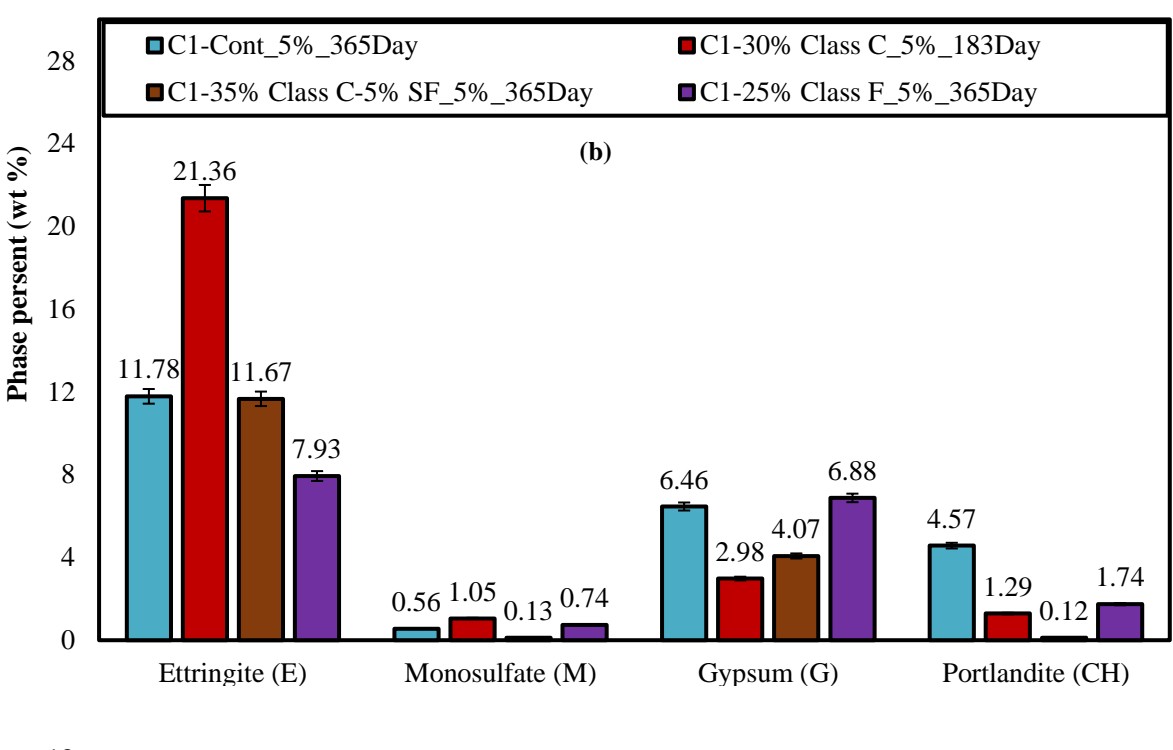

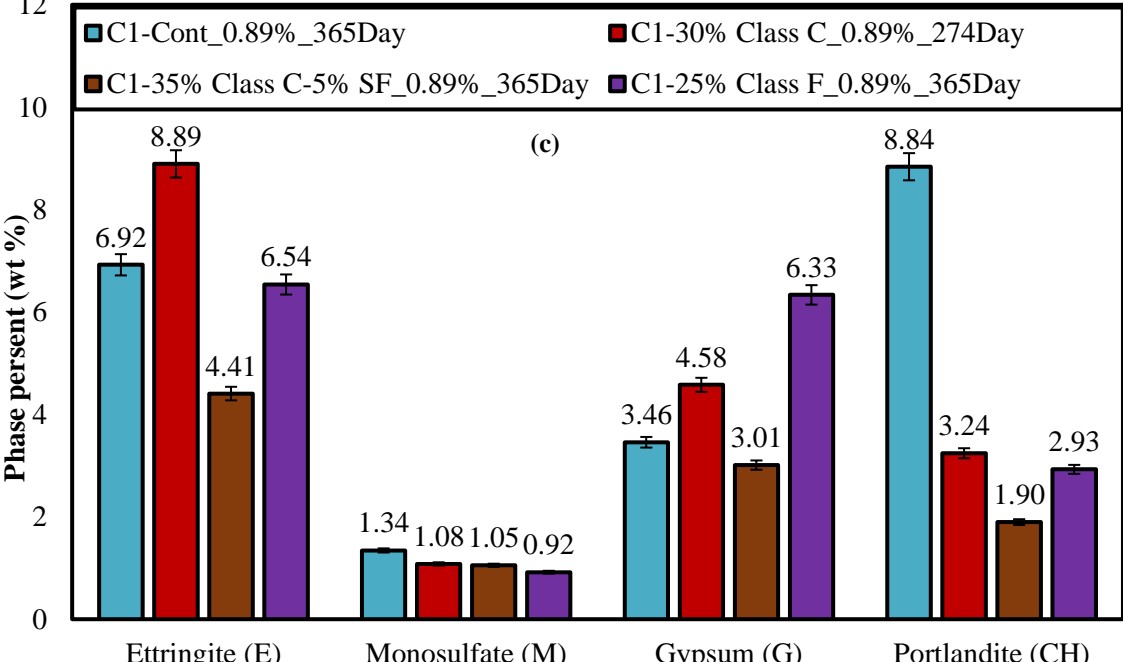

**Figure 11.** (**a**): Rietveld results for mortars using cement C1 submerged in 5% Na2SO4 after 56 days, (**b**): Rietveld results for mortars using cement C1 submerged in 5% $Na_2SO_4$ after 1 yr, (**c**): Rietveld results for mortars using cement C1 submerged in 0.89% $Na_2SO_4$ after 1yr.

## 5. Conclusions and Future Study Recommendation

In this study, the impact of exposure to two concentrations of sodium sulfate (e.g., 5 and 0.89%) on the specimens produced with binary and ternary binders supplied with high and low calcium fly ash has been investigated. The results can be outlined as the following:

- Exposure to sodium sulfate shows that the test specimens are damaged primarily by the formation and ongoing crystal growth pressure of ettringite independent of sodium

sulfate concentration. Microcracking was observed in the test samples, which led to an enhanced ingress of sulfate ions and consequently, accelerated the disintegration; however, the rate of attack is proportional to the concentration and supersaturation of the pores with sulfate ions.

- Gypsum was observed in very few instances when evaluated using the SEM independent of the solution concentration; however, traces of gypsum formation were evident in the XRD patterns and in relative amounts based on the Rietveld analysis of the diffraction patterns indicating that some deterioration some may have been attributed from gypsum formation.
- Significant differences in the mode of failure were evident between the two concentrations investigated. Larger cracks and in some cases warping and/or complete loss of cohesion were seen in samples placed in 5% sodium sulfate, especially high-calcium binary mixtures. Smaller cracks were observed in samples placed in 0.89% sodium sulfate with most deterioration only occurring at the ends and corners of the samples.
- SEM imaging revealed ettringite deposits were found within the paste matrix in both sodium sulfate concentrations; however, the higher concentration appeared more distinct and distributed throughout the matrix, while the amount and arrangement were discontinuous and significantly less dense in the lower concentration revealing little to no microcracking within the bulk paste matrix.

In the end, this study shows that regardless of sulfate concentration, ettringite is the main component to the expansion mechanism and ultimately leading the cementitious mixtures to deleterious sulfate attack. Additional research needs to be conducted on other mixtures including various cement types as well as the addition of other supplementary cementitious materials such as the blast furnace slag, silica fume, and other natural pozzolans. Further research should be conducted at even lower concentration of sulfate, such as 0.22% $Na_2SO_4$ (1500 ppm $SO_4^{2-}$) and 0.022% $Na_2SO_4$ (150 ppm $SO_4^{2-}$) corresponding to the class 1 and 2 potential exposure in ACI 201.2R. Additionally, more long-term sulfate attack durability tests, especially when samples are under actual loads need to be performed on concrete specimens submerged in various sulfate concentrations in the laboratory and benchmarked to actual field performance or the performance in outdoor sulfate exposure sites.

**Author Contributions:** Conceptualization, F.A. and M.N.; methodology, F.A.; software, F.A.; validation, F.A. and M.N.; formal analysis, F.A.; investigation, F.A.; resources, F.A.; data curation, F.A.; writing—original draft preparation, F.A. and M.N.; writing—review and editing, F.A. and M.N; visualization, F.A. and M.N; supervision, F.A.; project administration, F.A.; funding acquisition, F.A. All authors have read and agreed to the published version of the manuscript.

**Funding:** No funding was received for this study.

**Institutional Review Board Statement:** Not applicable.

**Informed Consent Statement:** Not applicable.

**Data Availability Statement:** Not applicable.

**Acknowledgments:** The authors acknowledge and appreciate all the institutions that supported this study.

**Conflicts of Interest:** The authors declare no conflict of interest.

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
