# Peer review of "Deterioration of Mortar Bars Using Binary and Ternary Mixtures Immersed in Sodium Sulfate Solutions"

_ceramics, doi:10.3390/ceramics5040071_

Round 1
Reviewer 1 Report
The reviewed article presents the results of the experiment consisting in placing mortar bars in SO4 solution in accordance with the ASM C1012 / 1012M standard.
The types of cement selected for the tests, as well as the concentration of SO4 solution, are correctly selected, they can properly illustrate the purpose of the tests.
The applied research methods (SEM, EDS, XRD) allow to properly carry out the tests and assess the causes of destruction / damage to mortar bars.
In the last paragraph on page 7, Authors describe the sulfate attack as a two-step process, without further explaining what it depends on. It seems that it is worth deepening this point - the elongation of the sample in the first stage is small, because the diffusion of sulfate ions occurs slowly in the non-cracked material, the gradual increase in cracking causes an increase in the diffusion coefficient, the formation of more ettringite grains and more and more cracks in the material. The sulfate attack can be said to be a self-accelerating process. For a good description of this, see Basista M, Weglewski W. Chemically Assisted Damage of Concrete: A Model of Expansion Under External Sulfate Attack. International Journal of Damage Mechanics. 2009; 18 (2): 155-175. doi: 10.1177 / 1056789508097540
The main conclusion of the work is the confirmation of the general opinion that the sulphate attack is mainly caused by ettringite crystallization in the pores, which leads to damage to the cement mortar.
As an extension of the presented research, the authors propose tests with the use of various cements and various sulphate concentrations. In my opinion, it would be interesting to test the sulphate attack on samples under load in order to bring the experimental conditions closer to the real load of the structure.
Reviewer 2 Report
This paper presents novel research on sodium sulfate attack to different kinds of mortars made using class C and class F fly ashes. Maybe the paper will better suit another journal since these are not ceramic materials, for the benefit of higher reading and citing. The experimental set-up was well-defined, and the text is clear and well-written.
Some minor changes that could be added to this manuscript follow:
11. The changes made to ASTM standardized testing of this kind should be highlighted in the abstract section.
22. The numbering of subheadings should be changed. Also, the referencing in the text should be done according to the instructions for the authors of this journal.
33. English should be improved in the section “Research significance”.
44. It would be beneficial to present a granulometric analysis of the materials used for the mixtures.
55. Fig. 7, please improve the title.
